# Synthesis, Properties, and Biological Activity Evaluation of Some Novel Naphtho[2,1-e]pyrazolo[5,1-c][1,2,4]triazines

**DOI:** 10.3390/ijms26167681

**Published:** 2025-08-08

**Authors:** Ion Burcă, Vasile-Nicolae Bercean, Gerlinde-Iuliana Rusu, Raluca Pop, Alexandra-Mihaela Diaconescu, Valentin Badea, Francisc Péter

**Affiliations:** 1Department of Applied Chemistry and Organic and Natural Compounds Engineering, Politehnica University Timisoara, Vasile Pârvan 6 Blvd., 300223 Timisoara, Romania; 2Faculty of Pharmacy, “Victor Babeș” University of Medicine and Pharmacy Timișoara, 2 Eftimie Murgu Square, 300041 Timișoara, Romania

**Keywords:** diazotization, diazo compound, cyclization, pyrazolo[5,1-c][1,2,4]triazines

## Abstract

This paper explores a synthetic pathway for naphtopyrazolotriazines utilizing amines as versatile starting materials. The approach leverages the reactivity of amines to construct the triazine core, fused with naphtho and pyrazolo cycles, through a series of controlled diazo coupling and cyclization reactions. By employing amines, this method allows for the introduction of varied substituents, enabling the tailoring of electronic and steric properties to suit specific potential applications. The significance of this work lies in its efficiency, scalability, and potential to synthesize compounds with tunable functionalities. Naphtopyrazolotriazines are of interest due to the presence of a pyrazolo triazine moiety, which is known for its bioactivity, including anticancer and antimicrobial properties, and their possible utility in optoelectronic materials. All synthesized compounds have been characterized by 1D and 2D NMR, IR, UV-Vis, and mass spectrometry. Additionally, UV-Vis and fluorescence spectra of the synthesized compounds, together with the frontier molecular orbitals energies, were calculated by DFT methods implemented in Gaussian 09W software.

## 1. Introduction

The synthesis and characterization of nitrogen-containing cyclic compounds represent an important research direction in heterocyclic chemistry. These compounds are of great interest because of their wide range of applications in pharmaceuticals, industrial chemicals, dyes, and materials science. A great number of natural compounds, such as alkaloids, nucleotides, and vitamins, contain nitrogen heterocycles in their structure. The presence of a nitrogen atom influences properties such as basicity, nucleophilicity, and the capability to form hydrogen bonds and metal complexes. Naphtho[2,1-e]pyrazolo[5,1-c][1,2,4]triazines represent a class of fused heterocycles, which are versatile compounds with numerous uses, mainly due to their extended electronic conjugation. Unlike pyrazolo triazines, they are much rarer compounds and do not have frequent mention in the literature.

The properties of naphtopyrazolo[5,1-c][1,2,4]triazines can be tailored according to a specific field of use by functionalization or by fusion of additional cycles. In this paper, a synthetic pathway for some new naphtho[2,1-e]pyrazolo[5,1-c][1,2,4]triazines is presented. Amines, specifically aminopyrazoles, were chosen as starting materials because they are versatile reagents used for the synthesis of a spectrum of fused and non-fused heterocyclic systems, many of which have uses in pharmaceutics and medicinal chemistry [1].

In a manner similar to that of primary aromatic amines, aminopyrazoles can be used in the synthesis of azo compounds by the well-known azo-coupling reaction of diazonium salts. Pyrazole-derived dyes generally have a good tinctorial strength and brighter dyeing than aniline-derived dyes [2]. Azo compounds containing the pyrazole moiety also have other different uses, including inhibition of enzyme activity [3], nonlinear optical applications [4], antimicrobial agents [5], synthesis of metal complexes with biological activities [6], and fluorescent probes for metal ion detection [7].

Some dyes containing pyrazoles can undergo intramolecular cyclization, resulting in fused heterocyclic systems containing the 1,2,4-triazine moiety [8]. An example of such compounds is naphtopyrazolotriazines, which are relatively small and poorly studied heterocyclic systems with many promising properties, for example, steroid-like, as they are related to azasteroids [9], antitumoral [10,11], antischistosomal [12,13] actions, and fluorescence [14].

This work presents the synthesis, characterization, and study of thermal and biological properties, as well as a computational study of several novel naphtho[2,1-e]pyrazolo[5,1-c][1,2,4]triazines.

## 2. Results

Diazonium salts (**2a**) and (**2b**), obtained by diazotization of the amine group of the heterocyclic amines ethyl 5-amino-3-methyl-1*H*-pyrazole-4-carboxylate (**1a**) and 3-phenyl-1*H*-pyrazol-5-amine (**1b**), were used in diazo coupling reactions with 2,7-naphthalenediol. Thus, the diazo compounds ethyl 5-((2,7-dihydroxynaphthalen-1-yl)diazenyl)-3-methyl-1*H*-pyrazole-4-carboxylate (**3a**) and 1-((3-phenyl-1*H*-pyrazol-5-yl)diazenyl)naphthalene-2,7-diol (**3b**) were formed. The overall reaction path is shown in Figure 1:

By further thermic cyclization of the compounds (**3a**) and (**3b**) in a suitable solvent, the following naphthopyrazolotriazines were obtained, ethyl 9-hydroxy-2-methylnaphtho[2,1-e]pyrazolo[5,1-c][1,2,4]triazine-1-carboxylate (4a) and 2-phenylnaphtho[2,1-e]pyrazolo[5,1-c][1,2,4]triazin-9-ol (**4b**), as seen in Figure 2:

The ester (**4a**) ethyl 9-hydroxy-2-methylnaphtho[2,1-e]pyrazolo[5,1-c][1,2,4]triazine-1-carboxylate was then hydrolyzed in an alkaline medium under reflux to produce 9-hydroxy-2-methylnaphtho[2,1-e]pyrazolo[5,1-c][1,2,4]triazine-1-carboxylic acid (5), which by further decarboxylation under reflux in a mixture of acetic and hydrochloric acid produced 2-methylnaphtho[2,1-e]pyrazolo[5,1-c][1,2,4]triazin-9-ol (**6**) (Figure 3):

The structure and purity of the synthesized compounds were confirmed by 1D and 2D NMR, IR spectroscopy, and LC-HRMS or HRMS.

All synthesized compounds were tested for potential biological activity against nine human pathogen microorganisms and one cancer cell line. Biological testing showed that none of the compounds obtained fell within the limits necessary to be declared to possess biological activity. Only compound (**3b**) was found to be active on the Hep-G2 cell line, indicating that it shows either anti-cancer activity or hepatic toxicity.

From thermal analysis data, it can be concluded that the cyclization is a purely thermal process, which leads to the loss of a water molecule and closure of the triazine cycle.

All of the mentioned compounds exhibit radiation absorption in the visible domain. The maximum absorbance occurs in the wavelength range of 400–463 nm.

Only naphtopyrazolotriazines (**4b**) and (**6**) exhibit fluorescence, with a maximum emission at the wavelength of 535 nm.

## 3. Discussion

The first mention of naphtopyrazolotriazines formation by thermic cyclization of diazo compounds derived from naphthalene and indazole is Bamberger’s work [15]. He studied the anhydrification of some diazo dyes derived from *β*-naphtol. Studies of 1,2,4-triazine by cyclization of pyrazole-derived azo dyes continued over the years [16,17].

According to the literature [9,18], the formation of the triazinic ring follows an intramolecular condensation reaction between the -OH group (attached to the C-7 atom of the naphthalenic cycle) and the pyrazolic -NH functional group with the elimination of a water molecule. For the compounds mentioned in the present work, the thermal cyclization reaction can be represented in the following scheme (Figure 4):

### 3.1. Thermal Analysis

Compounds (**3a**) and (**3b**) were analyzed by Differential Scanning Calorimetry (DSC) and Thermogravimetric (TG) analysis in order to study or prove the loss of a water molecule during the thermic cyclization. As can be seen in Figure 1, in the temperature range of 30–90 °C, an endothermic peak appears on the DSC curve along with a mass loss on the TG curve. This process can be associated with the evaporation of residual solvent. Furthermore, at approximately 100 °C on the DSC curve, a weak endothermic effect can be seen along with a small weight loss on the TG curve. This behavior can be explained by the intramolecular dehydration of the compound. This process is probably not complete because between 200 and 250 °C, another weight loss can be seen on the TG curve with an endothermic effect on the DSC curve, when the dehydration process is complete.

For compound (**3b**), from the thermogram (Figure 2), it can be seen that there is a mass loss step between 200 °C and 250 °C of approximately 6% (*w*/*w*), which corresponds to intramolecular elimination of water.

In conclusion, the thermal behavior of compounds (**3a**) and (**3b**) in the temperature range of 200–250 °C is similar, which confirms that the formation of the triazine ring follows the proposed reaction scheme.

### 3.2. Biological Activity Evaluation

Compounds (**3a**), (**3b**), (**4a**), (**4b**), (**5**), and (**6**) were part of the testing of a larger group of compounds tested against various human pathogens and HepG2 cells performed by EU-OPENSCREEN ERIC. Anti-fungal assays targeted *Candida albicans* ATCC 64124, *Aspergillus fumigatus* ATCC 46645, and *Candida auris* DSM21092, while anti-bacterial assays focused on *Enterococcus faecalis* ATCC 29212, *Staphylococcus aureus* MSSA ATCC 29213, *Pseudomonas aeruginosa*, *Escherichia coli* ATCC 25922, *Klebsiella pneumoniae* DSM681, and *Acinetobacter baumannii* DSM30007. These assays utilized 384-well plates with standardized microbial suspensions, incubated at 37 °C for 24 h, and measured growth inhibition by absorbance (or fluorescence for *A. fumigatus*) using an EnVision Multilabel Reader (PerkinElmer, Shelton, CT, USA) or Synergy HT Multi-Mode Reader (Agilent, Santa Clara, CA, USA). Activity was expressed as percentage growth inhibition, with thresholds that define compounds as active (≥70% or ≥50% depending on the assay), inconclusive (50–70%), or inactive (<50%). A cell viability ATP quantification assay on HepG2 cells used luminescence detection with a PHERAstar FS reader to assess cytotoxicity at 10 μM, with active compounds being defined as having ≥30% inhibition. Data processing involved Genedata Screener V18 software or normalization against controls, ensuring reproducibility and sensitivity. In the tables below, the inhibition percentages at a given molar concentration are given for each tested compound. Besides them, in the tables below are listed the hypothetical values of biological activities for known and widely used anti-bacterial, anti-fungal, and anti-cancer compounds calculated based on MIC (minimum inhibitory concentration for anti-bacterial and anti-fungal compounds) or IC50 (inhibitory concentration 50 for the anti-cancer compound). Preliminary results show that only compound (**3b**) was found to be biologically active, specifically, against Hep-G2 cancer cells (highlighted in grey) (Table 1, Table 2, Table 3 and Table 4).

### 3.3. Computational Studies

For the synthesized naphtho[2,1-e]pyrazolo[5,1-c][1,2,4]triazines, DFT and TD-DFT computations were performed at the B3LYP/6-311G level of theory with ethanol as solvent; the IEFPCM (Integral Equation Formalism Polarizable Continuum Model) solvation model was chosen for all computations. TD-DFT computations of the vertical excitations were performed; for each compound, a number of five excited states were considered.

HOMO orbitals are mainly located on the entire skeleton of the heterocycle and hydroxyl group (to a lesser extent on the pyrazole ring). In addition, it can be observed that in none of the naphtho[2,1-e]pyrazolo[5,1-c][1,2,4]triazines, the HOMO orbital does not appear in the substituents grafted onto the pyrazole moiety.

LUMO orbitals are mainly located on the entire skeleton of the heterocycle (to a lesser extent on the benzene ring that bears the hydroxyl group). In Table 5, the HOMO and LUMO orbitals of compounds (**3a**), (**3b**), (**4a**), (**4b**), (**5**), and (**6**) are represented:

On the basis of the frontier molecular orbital energies, the HOMO-LUMO gap and the values of some global reactivity descriptors have been computed (Table 6).

Although the HOMO-LUMO gaps are found within a small range of 0.031 eV and show no difference among the investigated compounds, the presence of the -COO^−^ group in compounds (**4a**) and (**5**) leads to slightly higher values of the HOMO and LUMO energies. This is reflected in the calculated values of the chemical potential and global electrophilicity, which present higher values for compounds (**4a**) and (**5**). The excitation energies and calculated wavelengths are depicted in the following tables (Table 7, Table 8, Table 9 and Table 10):

A molecular docking study, together with the calculation of the steric parameters of the synthesized compounds, has been performed.

The descriptors of the molecular shape [27], namely ovality, Connolly accessible area, Connolly solvent-excluded volume, and the partition coefficient logP, have been obtained with Chem3D V20.1 software. Autodock Vina [28] has been employed for the docking simulation. The structure of the receptor (PDB ID: 3GCW [29]) was downloaded from the Protein Data Bank [30]. The aforementioned structure refers to the protein structure of the epidermal growth factor-like domain of the low-density lipoprotein receptor in complex with PCSK9 (Proprotein Convertase Subtilisin/Kexin type 9). The HepG2 cell line, derived from human hepatocellular carcinoma, is frequently used in studies involving this interaction. A grid box of 40 × 40 × 40 Å was used, with the center of the grid box being considered the center of the protein. The optimized structures of the substituted porphyrins were loaded as ligands, and the torsions along the rotatable bonds were assigned. The visualization of the results was also performed by means of AutoDock Vina V1.2.7 software [28].

The results presented in Table 11 suggest that the steric parameters do not largely influence the affinity of the ligands towards the receptors. It can be observed that the compounds (**3b**) and (**4b**), the ones with the highest binding energies according to the molecular docking study, have similar steric properties to the other structures within the series that show lower binding affinities.

The ligand–receptor interactions, together with the ligand structure that led to the highest binding affinity for each compound, are presented in the Appendix A.

The binding affinities of the six investigated compounds towards the chosen receptor are depicted in Table 12 and outline that the best results have been obtained for compounds (**4b**) and (**3b**).

The average value of the binding affinity has been calculated, and the results are presented in Table 13.

The interactions between each ligand and receptor, which led to the results calculated in Table 12 and Table 13, are illustrated in Table 14. The figures corresponding to these interactions are included in the Appendix A.

According to the results presented in Table 14, the compounds (**3b**) and (**4b**) are characterized by a larger number of close-contact atom interactions; also, the presence of the phenyl group allows the interaction with the Asp651 residue, which do not appear for the other compounds (**3a**), (**4a**), (**5**) and (**6**).

## 4. Materials and Methods

The reagents were purchased from commercial sources and used as received. The 5-amino-3-methyl-1*H*-pyrazole-4-carboxylate hydrochloride was synthesized earlier in our laboratory following modified procedures from the literature [31,32]. 3-phenyl-1*H*-pyrazol-5-amine was also synthesized in our laboratory following modified procedures from the literature [33].

The 1D and 2D NMR spectra were recorded on a Bruker Avance III 500 MHz spectrometer (Bruker Daltonics, Billerica, MA, USA). Chemical shifts (δ) were measured in ppm and coupling constants (J) in Hz. TMS was used as an internal standard. The samples were dissolved in DMF-*d*7 or DMSO-*d*6.

IR spectra were recorded on a Jasco FT/IR-410 spectrophotometer (Jasco Corporation, Tokyo, Japan) in KBr pellets.

UV–Vis spectra were recorded on a Jasco V-530 spectrometer (Jasco Corporation, Tokyo, Japan), and the samples were dissolved in 96% ethanol. The concentrations of the analyzed compounds were as follows (Table 15):

Emission spectra were recorded on a Perkin Elmer LS 55 spectrophotometer (PerkinElmer, Shelton, CT, USA).

Melting points were measured on a Böetius PHMK apparatus (Veb Analytik, Dresden, Germany) and were not corrected.

Compounds were analyzed by LC-HRMS on an AGILENT 1290 LCTOF system (Agilent, Santa Clara, CA, USA) with the following parts: Multisampler G7167B, Flex pump G7104A, High speed pump G7120A, DAD WR G7115A, Isopump G7110B, TOF: G6230B.

The high-resolution MS (HRMS) spectrum was recorded on a Bruker Maxis II QTOF spectrometer (Bruker Daltonics, Bremen, Germany) with electrospray ionization (ESI) in positive mode. The compound was initially dissolved in DMSO and further diluted 1:100 with acetonitrile. MS spectrum processing and isotope pattern simulations have been performed using Compass Data Analysis V.4.4 (Bruker Daltonics, Billerica, MA, USA).

Thermogravimetric analysis was performed using the TG 209 F1 Libra from Netzsch (Selb, Germany). The sample was weighed in an open alumina crucible and heated at a rate of 1 K/min from 20 to 300 °C under a nitrogen atmosphere.

DSC analysis was performed using the DSC 204 Phonix instrument from Netzsch. The sample was weighed in a sealed alumina crucible and heated at a rate of 1 K/min from 20 to 300 °C under a nitrogen atmosphere.

The geometry optimization, vibrational analysis, and fluorescence properties of the heterocyclic compounds (**4a**), (**4b**), (**5**), and (**6**) were calculated using the G09W V16 software.

### Compounds’ Synthesis Procedure

The synthesis procedures for compounds (**2a**), (**2b**), (**3a**), (**3b**), (**4a**), (**4b**), (**5**), and (**6**) are presented below:4-(ethoxycarbonyl)-3-methyl-1*H*-pyrazole-5-diazonium chloride synthesis (**2a**)

In a beaker, ethyl 5-amino-3-methyl-1*H*-pyrazole-4-carboxylate hydrochloride (0.571 g; 2.75 mmol), synthesized earlier in our laboratory [17,18], was dissolved in 3.5 mL of hydrochloric acid solution (1:6 *v*/*v*) under stirring. An orange-yellow solution was formed, and the beaker was placed in an ice-water bath to cool the obtained solution. In parallel, a sodium nitrite solution was prepared by dissolving nitrite (0.201 g; 2.89 mmol) in 1 mL of distilled water. Both solutions were cooled in ice-water baths to ensure that the temperature was below 5 °C. When the solutions reached the necessary temperature, sodium nitrite solution was added dropwise, under stirring, to the acidic amine solution. An immediate color change was observed; i.e., the solution turned yellow. Stirring was continued for 30 min, while the temperature of the mixture was maintained below 5 °C by using an ice-water bath. The diazonium salt was not isolated and was used in further steps as a solution.

3-phenyl-1*H*-pyrazole-5-diazonium chloride synthesis (**2b**)

In a beaker, 3-phenyl-1*H*-pyrazol-5-amine (0.852 g; 5.25 mmol) was dissolved in 3 mL of hydrochloric acid in an acetic acid solution (1:3 *v*/*v*) under vigorous stirring. An orange solution was formed, and the beaker was placed in an ice-water bath to cool the obtained solution. In parallel, a sodium nitrite solution was prepared by dissolving nitrite (0.372 g; 5.33 mmol) in 2 mL of distilled water. Both solutions were cooled in ice-water baths to ensure that the temperature was below 5 °C. When the solutions reached the necessary temperature, sodium nitrite solution was added dropwise, under stirring, to the acidic amine solution. After several minutes, the diazonium salt precipitated from the solution. Stirring was continued for 30 min, while the temperature of the mixture was maintained below 5 °C by using an ice-water bath. Diazonium salt was separated from the mixture by vacuum filtration, but it was not fully dried.

Ethyl 5-((2,7-dihydroxynaphthalen-1-yl)diazenyl)-3-methyl-1*H*-pyrazole-4-carboxylate (**3a**)

In a round bottom flask equipped with a magnetic stirrer and a thermometer, 2,7-naphthalenediol (0.413 g; 2.5 mmol) and sodium hydroxide (0.200 g; 5 mmol), along with 20 mL of 96% ethanol and 20 mL of pyridine and were added and stirred. Once the solids were fully dissolved, the flask was placed in an ice-water bath to maintain the temperature below 5 °C.

After both solutions (**2a** and 2,7-naphthalenediol solution) were cooled, diazonium chloride solution was added dropwise to the 2,7-naphthalenediol solution, with the two solutions being vigorously stirred while cooling was maintained. An intense orange coloration of the resulting solution was observed. Using a 30% sodium hydroxide solution, the pH was adjusted to 7–8. The cooling was maintained for 3.5 h. Then, the reaction mass was stirred for 24 h at room temperature.

After 24 h, the contents of the flask were acidified using 99.8% acetic acid until reaching a mildly acidic pH of 5.5–6. The dark orange suspension was then poured into a beaker that contained 200 mL of distilled water in order to ensure complete precipitation of the azo dye. The precipitate was then separated via vacuum filtration. The purity of the product was high enough to use it as is in the further synthesis steps.

Yield: 87%, m.p.: >300 °C (thermal cyclization occurs at temperatures >100 °C); ^1^H NMR: **δ_H_** (DMF-*d*_7_, 500 MHz, ppm): 16.16 (s, 1H, 2-C-OH), 10.51 (s, 1H, 7-C-OH), 7.93 (d, 1H, *J* = 1.88 Hz, 8-H), 7.83 (d, 1H, *J* = 9.56 Hz, 4-H), 7.56 (d, 1H, *J* = 8.44 Hz, 5-H), 7.00 (dd, 1H, *J*_1_ = 8.39 Hz, *J*_2_ = 2.52 Hz, 6-H), 6.51 (d, 1H, *J* = 9.56 Hz, 3-H), 4.44 (q, 2H, *J* = 7.16 Hz, -CH_2_-CH_3_), 2.54 (s, 3H, -CH_3_), 1.46 (t, 3H, *J* = 7.16 Hz, -CH_2_-CH_3_); ^13^C NMR: **δ_C_** (DMF-*d*_7_, 125 MHz, ppm): 179.1 (2-C), 163.8 (-C=O), 160.3 (7-C), 152.3 (5′-C), 146.8 (3′-C), 143.5 (4-C), 136.4 (9-C), 131.84 (5-C), 131.8 (1-C), 122.9 (3-C), 122.4 (10-C), 116.6 (6-C), 108.5 (8-C), 99.7 (4′-C), 60.8 (O-CH_2_-CH_3_), 14.7 (O-CH_2_-CH_3_), 12.1 (-CH_3_); FT-IR (cm^−1^): 3453 (ν_N-H_), 3061 (ν_Car-H_), 1664 (ν_C=O_), 1496, 1436 (δ_C-H_), 1017, 885 (γ_Car.-H_); UV-Vis (96% ethanol): λ_max_ = 463 nm, ε(λ_max_) = 18,211 M^−1^ cm^−1^; LC-HRMS (*m*/*z*): [M+H]^+^ for C_17_H_16_N_4_O_4_ calcd. 341.1244, found 341.1237, [M+Na]^+^ for C_17_H_16_N_4_O_4_ calcd. 363.1064, found 363.1061.

All spectra are reported in the Appendix A.

1-((3-phenyl-1*H*-pyrazol-5-yl)diazenyl)naphthalene-2,7-diol synthesis (**3b**)

In a round bottom flask equipped with a magnetic stirrer and a thermometer, 2,7-naphthalenediol (0.788 g; 4.78 mmol) and sodium hydroxide (0.210 g; 5.01 mmol), along with 20 mL of 96% ethanol and 40 mL of pyridine and were added and stirred. Once the solids were fully dissolved, the flask was placed in an ice-water bath to maintain the temperature below 5 °C.

After the 2,7-naphthalenediol solution was cooled, diazonium chloride precipitate was added in small portions to the 2,7-naphthalenediol solution under vigorous stirring and cooling. A blood-red coloration of the resulting solution was observed. Using a 30% sodium hydroxide solution, the pH was adjusted to 7–8. Cooling and stirring were maintained for 5 h.

After 5 h, the contents of the flask were acidified using 99.8% acetic acid until reaching a mildly acidic pH of 5.5–6. The dark orange suspension was then poured into a beaker that contained 450 mL of distilled water in order to ensure complete precipitation of the azo dye. The precipitate was then separated via vacuum filtration. The purity of the product was high enough to use it as is in the further synthesis steps.

Yield: 33%, m.p.: >300 °C (thermal cyclization occurs at temperatures >100 °C); ^1^H NMR: **δ_H_** (DMSO-*d*6, 500 MHz, ppm): 15.14 (s, 1H, 2-C-OH), 13.83 (s, 1H, 1′-NH), 10.09 (s, 1H, 7-C-OH), 8.02 (d, 1H, *J* = 1.66 Hz, 8-H), 7.92–7.88 (m, 3H, 4-H, 2′′-H, 6′′-H), 7.73 (d, 1H, *J* = 8.51 Hz, 5-H), 7.56–7.53 (m, 2H, 3″-H, 5′′-H), 7.45 (t, 1H, *J* = 7.37 Hz, 4′′-H), 7.19 (s, 1H, 4′-H), 7.02 (dd, 1H, *J*_1_ = 8.55 Hz, *J*_2_ = 2.06 Hz, 6-H), 6.88 (d, 1H, *J* = 9.05 Hz, 3-H); ^13^C NMR: **δ_C_** (DMSO-6, 125 MHz, ppm): 160.1 (5′-C), 159.5 (2-C), 158.0 (7-C), 144.1 (3′-C), 137.4 (4-C), 134.5 (10-C), 130.4 (5-C), 129.0 (3′′-C, 5′′-C), 128.7 (4′′-C), 128.5 (1′′-C), 125.3 (2′′-C, 6′′-C), 122.1 (9-C), 117.2 (3-C), 116.0 (6-C), 104.4 (8-C), 91.1 (4′-C); ^15^N NMR **δ_N_** (DMSO-*d*6, 50.66 MHz, ppm): 294.3 (2′-N), 199.9 (1′-N), 46.4 (1-C-N = N); FT-IR (cm^−1^): 3052 (ν_Car-H_), 1743 (ν_C=O_), 1621 (ν_Car_), 1473, 1410 (δ_C-H_), 758 (γ_Car.-H_); UV-Vis (96% ethanol): λ_1_ = 412.5, λ_max_ = 453 nm, ε(λ_max_) = 15,975 M^−1^ cm^−1^; LC-HRMS (*m*/*z*): [M+H]^+^ for C_19_H_14_N_4_O_2_ calcd. 331.1190, found 331.1185, [M+Na]^+^ for C_19_H_14_N_4_O_2_ calcd. 353.1009, found 353.1026.

All spectra are reported in the Appendix A.

Ethyl 9-hydroxy-2-methylnaphtho[2,1-e]pyrazolo[5,1-c][1,2,4]triazine-1-carboxylate (**4a**)

In a round bottom flask equipped with a magnetic stirrer, 18.5 mL of cyclohexanol and **3a** compound (0.340 g; 1 mmol) were added under stirring. The round bottom flask was equipped with a reflux refrigerator, and the heating and stirring were turned on. After approximately 20 min, the solid was fully dissolved, and a bright orange color was observed. The heating and stirring continued for 15 more hours, after which a dark yellow solid precipitated from the solution. The solvent was evaporated under vacuum. After separation, the precipitate was washed several times with 96% ethanol and water.

Yield: 86%, m.p.: >300 °C; ^1^H NMR: **δ_H_** (DMSO-*d*6, 500 MHz, ppm): 10.67 (s, 1H, 7-C-OH), 8.52 (d, 1H, *J* = 2.39 Hz, 8-H), 8.33 (d, 1H, *J* = 8.80 Hz, 3-H), 7.99 (d, 1H, *J* = 8.80 Hz, 4-H), 7.93 (d, 1H, *J* = 8.80 Hz, 5-H), 7.29 (dd, 1H, *J*_1_ = 2.39 Hz, *J*_2_ = 8.80 Hz, 6-H), 4.37 (q, 2H, *J* = 7.15 Hz, O-CH_2_-CH_3_), 2.65 (s, 3H, -CH_3_), 1.41 (t, 3H, O-CH_2_-CH_3_); ^13^C NMR: **δ_C_** (DMSO-6, 125 MHz, ppm): 162.0 (-C=O), 159.3 (7-C), 156.5 (3′-C), 145.5 (5′-C), 138.3 (3-C), 134.2 (9-C), 131.17 (5-C), 131.14 (2-C), 125.0 (10-C), 123.7 (1-C), 119.4 (6-C), 107.7 (4-C), 105.3 (8-C), 102.4 (4′-C), 59.9 (O-CH_2_-CH_3_), 14.5 (-CH_3_), 14.3 (O-CH_2_-CH_3_); ^15^N NMR **δ_N_** (DMSO-*d*6, 50.66 MHz, ppm): 272.7 (2′-N), 205.7 (1′-N); FT-IR (cm^−1^): 3099 (ν_Car-H_), 2979 (ν_Calk-H_), 1707 (ν_C=O_), 1472, 1410 (δ_C-H_), 1074 (ν_C-O_), 884, 782 (γ_Car.-H_); UV-Vis (96% ethanol): λ_max_ = 420.5 nm, ε(λ_max_) = 15,307 M^−1^ cm^−1^; LC-HRMS (*m*/*z*): [M+H]^+^ for C_17_H_14_N_4_O_3_ calcd. 323.1139, found 323.1136, [M+Na]^+^ for C_17_H_14_N_4_O_3_ calcd. 345.0958, found 345.0948.

All spectra are reported in the Appendix A.

2-phenylnaphtho[2,1-e]pyrazolo[5,1-c][1,2,4]triazin-9-ol (**4b**)

In a round bottom flask equipped with a magnetic stirrer, a solution composed of 10 mL of concentrated hydrochloric acid, 15 mL of glacial acetic acid, and **3a** compound (0.340 g; 1mmol) was added under stirring. The round bottom flask was equipped with a reflux refrigerator, and the heating and stirring were turned on. The heating and stirring were continued for 15 more hours, after which the solution was alkalinized using 30% sodium hydroxide to a pH of 6–7. Then, the reaction mixture was poured over 100 mL of distilled water. A dark pearlescent orange precipitate was formed, and it was washed several times with 96% ethanol and water.

Yield: 59%, m.p.: >300 °C; ^1^H NMR: **δ_H_** (DMSO-*d*6, 500 MHz, ppm): 10.62 (s, 1H, 7-C-OH), 8.66 (d, 1H, *J* = 2.14 Hz, 8-H), 8.46 (d, 1H, *J* = 8.85 Hz, 3-H), 8.29 (d, 1H, *J* = 8.85 Hz, 4-H), 8.23 (d, 2H, *J* = 7.38 Hz, 2″-H, 3″-H), 8.10 (m, 2H, 5-H, 4′-H), 7.57 (t, 2H, *J* = 7.40 Hz, 3″-H, 5″-H), 7.50 (t, 1H, *J* = 7.40 Hz, 4″-H), 7.33 (dd, 1H, *J*_1_ = 2.17 Hz, *J*_2_ = 8.60 Hz, 6-H); ^13^C NMR: **δ_C_** (DMSO-6, 125 MHz, ppm): 159.1 (7-C), 154.5 (3′-C), 148.8 (5′-C), 137.4 (3-C), 133.0 (9-C), 131.8 (2-C), 131.7 (1″-C), 131.2 (5-C), 129.5 (4″-C), 129.0 (3″-C, 5″-C), 126.4 (2″-C, 6″-C), 125.1 (10-C), 124.7 (1-C), 119.3 (6-C), 108.2 (4-C), 105.3 (8-C), 97.3 (4′-C); ^15^N NMR **δ_N_** (DMSO-*d*6, 50.66 MHz, ppm): 207.3 (2′-N); FT-IR (cm^−1^): 3138 (ν_Car-H_), 1618 (ν_Car_), 1458, 1422 (δ_C-H_), 1016 (δ_Car-H_), 893, 767 (γ_Car.-H_); UV-Vis (96% ethanol): λ_max_ = 400 nm, ε(λ_max_) = 7436 M^−1^ cm^−1^; Emission spectrum: λ_max_ = 535 nm; LC-HRMS (*m*/*z*): [M+H]^+^ for C_19_H_12_N_4_O calcd. 313.1084, found 313.1082, [M+Na]^+^ for C_19_H_12_N_4_O calcd. 335.0903, found 335.0886.

All spectra are reported in the Appendix A.

9-hydroxy-2-methylnaphtho[2,1-e]pyrazolo[5,1-c][1,2,4]triazine-1-carboxylic acid (**5**)

In a round bottom flask equipped with a magnetic stirrer and a reflux condenser, 24 mL of 10% sodium hydroxide was added together with compound (**4a**) (0.68 g; 2 mmol). The heating and stirring were turned on, and after the complete dissolution of the compound, the solution turned intense red-violet. After 20 h of reflux, the reaction mixture was left to cool down to room temperature. Then, 30 mL of distilled water was added to it. Glacial acetic acid was used to neutralize the reaction mixture. After neutralization, a yellow-brown solid was formed, and it was separated from the reaction mixture by vacuum filtration.

Yield: 39%, m.p.: >300 °C; ^1^H NMR: **δ_H_** (DMSO-d6, 500 MHz, ppm): 11.48 (s, l, 1H, 7-C-OH), 8.62 (s, 1H, 8-H), 8.39 (d, 1H, *J* = 8.67 Hz, 3-H), 8.06–8.02 (m, 2H, 4-H, 5-H), 7.31 (d, 1H, *J* = 7.39 Hz, 6-H), 2.74 (s, 3H, -CH_3_); ^13^C NMR: **δ_C_** (DMSO-d6, 125 MHz, ppm): 163.6 (-C=O), 159.3 (7-C), 156.6 (3′-C), 145.8 (5′-C), 138.2 (3-C), 134.1 (9-C), 131.22 (5-C), 131.18 (2-C), 124.9 (10-C), 123.8 (1-C), 119.4 (6-C), 107.9 (4-C), 105.2 (8-C), 103.5 (4′-C), 14.5 (-CH_3_); ^15^N NMR **δ_N_** (DMSO-d6, 50.66 MHz, ppm): 272.5 (2′-N), 205.5 (1′-N); FT-IR (cm^−1^): 3054 (ν_Car-H_), 1616 (ν_Car_), 1398 (δ_C-H_), 1241, 1137, 1018 (ν_C-O_), 895, 837 (γ_Car.-H_); UV-Vis (96% ethanol): λ_max_ = 400nm, ε(λ_max_) = 6487 M^−1^ cm^−1^; HRMS-ESI (*m*/*z*): [M+H]^+^ for C_15_H_10_N_4_O_3_ calcd. 295.0826, found 295.2594.

All spectra are reported in the Appendix A.

2-methylnaphtho[2,1-e]pyrazolo[5,1-c][1,2,4]triazin-9-ol (**6**)

In a round bottom flask equipped with a magnetic stirrer, a solution composed of 60 mL of concentrated hydrochloric acid and 130 mL of glacial acetic acid, and compound **5** (0.585 g; 2mmol) was added under stirring. The round bottom flask was equipped with a reflux refrigerator, and the heating and stirring were turned on. The heating and stirring were continued for 30 more hours, after which the solution was alkalinized using 30% sodium hydroxide to a pH of 6–7. Then, the reaction mixture was poured over 100 mL of distilled water. A dark orange precipitate was formed, and it was washed several times with 96% ethanol and water.

Yield: 75%, m.p.: >300 °C; ^1^H NMR: **δ_H_** (DMSO-d6, 500 MHz, ppm): 10.58 (s, 1H, 7-C-OH), 8.62–8.63 (d, 1H, *J* = 1.9 Hz, 8-H), 8.40 (d, 1H, *J* = 8.88 Hz, 3-H), 8.12 (d, 1H, *J* = 8.88 Hz, 4-H), 8.06 (d, 1H, *J* = 8.80 Hz, 5-H), 7.33 (s, 1H, 4′-H), 7.31 (d, 1H, *J* = 8.88 Hz, 6-H), 2.60 (s, 3H, -CH_3_); ^13^C NMR: **δ_C_** (DMSO-d6, 125 MHz, ppm): 159.0 (7-C), 153.9 (3′-C),148.2 (5′-C), 137.2 (3-C), 132.5 (9-C), 131.7 (2-C), 131.1 (5-C), 124.8 (10-C), 124.4 (1-C), 119.1 (6-C), 108.0 (4-C), 105.2 (8-C), 99.7 (4′-C), 14.2 (-CH_3_); ^15^N NMR **δ_N_** (DMSO-d6, 50.66 MHz, ppm): 274.4 (1′-N), 205.3 (2′-N); FT-IR (cm^−1^): 3049 (ν_Car-H_), 1614 (ν_Car_), 1450 (δ_C-H_), 805, 728 (γ_Car.-H_); UV-Vis (96% ethanol): λ_max_ = 400nm, ε(λ_max_) = 7436 M^−1^ cm^−1^; Emission spectrum: λ_max_ = 535.5nm; LC-HRMS (*m*/*z*): [M+H]^+^ for C_14_H_10_N_4_O calcd. 251.0927, found 251.0923, [M+Na]^+^ for C_14_H_10_N_4_O calcd. 273.0747, found 273.0736.

All spectra are reported in the Appendix A.

## 5. Conclusions

The current study represents a continuation of research on heterocyclic azo compounds [34,35] and naphtopyrazolotriazines [36] conducted by our research group. A total of six new compounds, which are not mentioned in the literature, were synthesized. Full characterizations of the compounds are provided.

It was proved that pyrazolic azo compounds derived from 2,7-naphtalenediol react in the same way described in the literature as similar compounds derived from β-naphtol; specifically, they undergo a thermal cyclization reaction along with a water molecule loss, after which fused heterocyclic compounds are formed. Some of the naphtopyrazolotriazines also exhibit fluorescence properties, which is also mentioned in the literature for similar compounds.

Diazo compounds (**3a**) and (**3b**) were analyzed by UV-Vis spectroscopy, but their molar absorption coefficients do not have high values; thus, these compounds are not suitable for use as azo dyes.

The property of gaining fluorescence by cyclization at raised temperatures can suggest a possible use of the synthesized or similar azo compounds as thermal chemosensors.

Although the docking studies showed that compounds (**3b**) and (**4b**) have the highest activity against Hep-G2 cells, the biological activity evaluation revealed promising results only for compound (**3b**). Further research in this direction may lead to important results in anti-cancer drug research.

## Data Availability

The data presented in this study are available within the article or Appendix A.

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
