# Peer review of "Synthesis, Properties, and Biological Activity Evaluation of Some Novel Naphtho[2,1-e]pyrazolo[5,1-c][1,2,4]triazines"

_ijms, 2025, doi:10.3390/ijms26167681_

Round 1
Reviewer 1 Report
Comments and Suggestions for Authors
Quntum-chemical calculations are performed correctly : geomentry optimization, electron transitions. In would be interest to show the shape of next orbitals; HOHO-2 and HOMO-6 … are lone electron pair (LEP) located at sp 2 -hybridized nitrogen atoms; this LEPs are acceptor centre of hydrogen bond between substrat and bio-mo0lecules and hence are conncted with biological aqctivity.
Also, it is to be intrest to perform the geometry optimization in the excited state and to estimate the first electron transition, corresponding to fluorescence. Stock’s shift can point on the geometrical changes upon excitation.
Therefore, these interesting investigations could be continued, for example, in next paper?
– In Schemes 1 - 4, it is necessary to indicate the complete reaction conditions (temperature, solvent etc.).
– In Table 1, it is necessary to add 2-3 antifungal and antibacterial drugs as a reference.
– Table 1: I believe that anticancer studies on the Hep G2 cell line should be presented in a separate table and an anticancer drug should be added for comparison, so that it would be possible to assess the anticancer activity of the compounds being studied adequately.
– Table 1. What do negative % inhibition values mean for all compounds?
– According to the authors, what is the reason for the maximum anticancer effect of the compound 5 (9-hydroxy-2-methylnaphtho[2,1-e]pyrazolo[5,1-c][1,2,4]triazine-1-carboxylic acid) on Hep G2 cells of hepatocellular carcinoma?
– Compound 4a. Line 323. The given calculated and experimental data correspond to the formula: [M+Na]+ for C17H14N4O3 calcd. 345.0958, found 345.0948.
They do not correspond to the erroneous formula: [M+Na] for C19H14N4O2 calcd. 345.0958, found 345.0948.
– Compound 4b. Line 344. The given calculated and experimental data correspond to the formula: [M+Na]+ for C19H12N4O calcd. 335.0903, found 335.0886.
They do not correspond to the erroneous formula: [M+Na] for C19H14N4O2 calcd. 335.0903, found 335.0886.
– Compound 5. Lines 362-363. The calculated and experimental data given correspond to the ion [M-H] - for C15H10N4O3 calcd. 293.0675, found 293.1744. They do not correspond to the ion [M+Na] for C15H10N4O3 calcd. 293.0675, found 293.1744.
– Compound 6. Lines 382-383. The calculated and experimental data given correspond to the formula: [M+Na] for C14H10N4O, calcd. 273.0747, found 273.0736. They do not correspond to the erroneous formula: [M+Na] for C19H14N4O2 calcd. 273.0747, found 273.0736.
The authors of the work should once again carefully check the correctness of the presentation of the experimental data.
Author Response
Quntum-chemical calculations are performed correctly : geomentry optimization, electron transitions. In would be interest to show the shape of next orbitals; HOHO-2 and HOMO-6 … are lone electron pair (LEP) located at sp 2 -hybridized nitrogen atoms; this LEPs are acceptor centre of hydrogen bond between substrat and bio-mo0lecules and hence are conncted with biological aqctivity.
Also, it is to be intrest to perform the geometry optimization in the excited state and to estimate the first electron transition, corresponding to fluorescence. Stock’s shift can point on the geometrical changes upon excitation.
Therefore, these interesting investigations could be continued, for example, in next paper?
Authors’ feedback: Thank you for suggestions, we may include them in our next papers. For now, a molecular docking study, together with the calculation of the steric parameters of the synthesized compounds have been added to the original submitted paper.
– In Schemes 1 - 4, it is necessary to indicate the complete reaction conditions (temperature, solvent etc.).
Authors’ feedback: Schemes 1 – 4 have been improved taking into account the Reviewer’s suggestion.
– In Table 1, it is necessary to add 2-3 antifungal and antibacterial drugs as a reference.
Authors’ feedback: Microorganisms were sorted out in separate tables (Gram positive/negative, fungi) for each category a reference drug activity is presented.
– Table 1: I believe that anticancer studies on the Hep G2 cell line should be presented in a separate table and an anticancer drug should be added for comparison, so that it would be possible to assess the anticancer activity of the compounds being studied adequately.
Authors’ feedback: Reference anti-cancer drug activity added.
– Table 1. What do negative % inhibition values mean for all compounds?
Authors’ feedback: negative values indicate that the compound not only failed to inhibit the growth of the target microorganism or cell line but may have even promoted growth.
– According to the authors, what is the reason for the maximum anticancer effect of the compound 5 (9-hydroxy-2-methylnaphtho[2,1-e]pyrazolo[5,1-c][1,2,4]triazine-1-carboxylic acid) on Hep G2 cells of hepatocellular carcinoma?
Authors’ feedback: According to the presented data, the compound 3b [1-((3-phenyl-1H-pyrazol-5-yl)diazenyl)naphthalene-2,7-diol] has the highest activity on Hep G2 cells. The reason for this is presented in the newly provided molecular docking study.
– Compound 4a. Line 323. The given calculated and experimental data correspond to the formula: [M+Na]+ for C17H14N4O3 calcd. 345.0958, found 345.0948.
They do not correspond to the erroneous formula: [M+Na] for C19H14N4O2 calcd. 345.0958, found 345.0948.
– Compound 4b. Line 344. The given calculated and experimental data correspond to the formula: [M+Na]+ for C19H12N4O calcd. 335.0903, found 335.0886.
They do not correspond to the erroneous formula: [M+Na] for C19H14N4O2 calcd. 335.0903, found 335.0886.
– Compound 5. Lines 362-363. The calculated and experimental data given correspond to the ion [M-H] - for C15H10N4O3 calcd. 293.0675, found 293.1744. They do not correspond to the ion [M+Na] for C15H10N4O3 calcd. 293.0675, found 293.1744.
– Compound 6. Lines 382-383. The calculated and experimental data given correspond to the formula: [M+Na] for C14H10N4O, calcd. 273.0747, found 273.0736. They do not correspond to the erroneous formula: [M+Na] for C19H14N4O2 calcd. 273.0747, found 273.0736.
Authors’ feedback: HRMS data has been reviewed and updated.
The authors of the work should once again carefully check the correctness of the presentation of the experimental data.

Reviewer 2 Report
Comments and Suggestions for Authors
The work is methodologically sound, and the compounds are well-characterised. The DFT analysis is appropriate for frontier orbital and electronic transition interpretation. The manuscript may be accepted after minor revision:
- Only compound 3b shows significant activity against Hep-G2 cells (62.6%). The authors should discuss why only 3b is active, potentially in a structure–activity relationship (SAR).
- The screening data against microbial strains showed predominantly low or negative inhibition values. It is recommended to explicitly state that most of the compounds lack antibacterial activity at 50 µM and to rationalise this in terms of compound structure or permeability.
- Emission spectra for only two compounds are shown (4b and 6). Are other compounds non-emissive? Please clarify this explicitly.
- Typos: "loose" (line 392) → should be "loss" of water. Inconsistent spacing around units: e.g., "50.0 μM/ 4.55" vs. "10000.0 nM/3.5h". Ensure a space after units and consistently use % inhibition/growth inhibition.
Author Response
- Only compound 3b shows significant activity against Hep-G2 cells (62.6%). The authors should discuss why only 3b is active, potentially in a structure–activity relationship (SAR).
Authors’ feedback: A molecular docking study, together with the calculation of the steric parameters of the synthesized compounds is provided in the updated manuscript.
- The screening data against microbial strains showed predominantly low or negative inhibition values. It is recommended to explicitly state that most of the compounds lack antibacterial activity at 50 µM and to rationalise this in terms of compound structure or permeability.
Authors’ feedback: Additional explanations are provided in the newly provided molecular docking study.
- Emission spectra for only two compounds are shown (4b and 6). Are other compounds non-emissive? Please clarify this explicitly.
Authors’ feedback: Only compounds 4b and 6 are emissive. The statement was added to the manuscript.
- Typos: "loose" (line 392) → should be "loss" of water. Inconsistent spacing around units: e.g., "50.0 μM/ 4.55" vs. "10000.0 nM/3.5h". Ensure a space after units and consistently use % inhibition/growth inhibition.
Authors’ feedback: Manuscript updated according to Reviewer’s suggestion.
